

# Xylose fermentation to ethanol by new *Galactomyces geotrichum* and *Candida akabanensis* strains

Raquel V. Valinhas[1], Lílian A. Pantoja[2], Ana Carolina F. Maia[3], Maria Gabriela C.P. Miguel[4], Ana Paula F.C. Vanzela[3], David L. Nelson[1] and Alexandre S. Santos[5]

[1] Graduate Program of Biofuels, Universidade Federal dos Vales do Jequitinhonha e Mucuri, Diamantina, Minas Gerais, Brazil
[2] Science and Technology Institute, Universidade Federal dos Vales do Jequitinhonha e Mucuri, Diamantina, Minas Gerais, Brazil
[3] Department of Pharmacy, Universidade Federal dos Vales do Jequitinhonha e Mucuri, Diamantina, Minas Gerais, Brazil
[4] Department of Biology, Universidade Federal de Lavras, Lavras, Minas Gerais, Brazil
[5] Department of Basic Sciences, Universidade Federal dos Vales do Jequitinhonha e Mucuri, Diamantina, Minas Gerais, Brazil

## ABSTRACT

The conversion of pentoses into ethanol remains a challenge and could increase the supply of second-generation biofuels. This study sought to isolate naturally occurring yeasts from plant biomass and determine their capabilities for transforming xylose into ethanol. Three yeast strains with the ability to ferment xylose were isolated from pepper, tomato and sugarcane bagasse. The strains selected were characterized by morphological and auxanographic assays, and they were identified by homology analysis of 5.8 S and 26 S ribosomal RNA gene sequences. The identities of two lineages of microrganism were associated with *Galactomyces geotrichum*, and the other was associated with *Candida akabanensis*. Fermentative processes were conducted with liquid media containing only xylose as the carbon source. $Y_{P/S}$ values for the production of ethanol ranging between 0.29 and 0.35 g g$^{-1}$ were observed under non-optimized conditions.

Corresponding author
Alexandre S. Santos,
alexandre.soares@ufvjm.edu.br,
alexandreletam@gmail.com

## INTRODUCTION

The lignocellulosic biomass is considered to be the most accessible and abundant renewable raw material existing on the planet (*Zhou et al., 2011*). The prevalent polysaccharide in the plant cell walls is cellulose, making up 40.6–51.2% of the wall material. The hemicelluloses comprise the other polysaccharidic fraction, representing 28.5–37.2% of the plant cell wall (*Pauly & Keegstra, 2008*). The carbohydrates present in plant cell walls could be transformed into ethanol by a technological route that consists of pretreatment of the lignocellulosic material, hydrolysis of polysaccharides and conversion of the sugars released into alcohol by a fermentative process (*Talebnia, Karakashev & Angelidaki, 2010*; *Bhatia, Johri & Ahmad, 2012*). During the pretreatment of the lignocellulosic biomass for the

production of cellulosic ethanol, the hydrolysis of the hemicellulose glycosidic bonds and, consequently, the release of the monosaccharides occur. Xylose is the most abundant monosaccharide resulting from the deconstruction of hemicellulose (*Gírio et al., 2010*). However, xylose and other pentoses released after the pretreatment of lignocellulosic material are frequently discarded because the microorganisms conventionally used in industry have no capacity to ferment pentoses (*Gírio et al., 2010*; *Claassen et al., 1999*). The conversion of the hemicellulose fraction from the lignocellulosic biomass into ethanol could represent an increase of 50% in the production of second-generation ethanol (*Nogué & Karhumaa, 2015*).

The selection of microorganisms with the ability to ferment pentoses is a strategy for improving the efficiency of the industrial use of lignocellulosic biomass (*Agbogbo & Coward-Kelly, 2008*; *Rao, Bhadra & Shivaji, 2008*; *Arora et al., 2015*; *Silva et al., 2016*). Even engineering xylose-fermenting microorganisms are constructed based on metabolic models and enzyme genes originally found in wild microorganisms, and many metabolic bottlenecks remain to be solved (*Moysés et al., 2016*). However, the ability to ferment both pentoses and hexoses is not widespread among microorganisms, and this is an obstacle for the efficient industrial production of second generation ethanol (*Talebnia, Karakashev & Angelidaki, 2010*). There are some yeast species that have already been identified as being capable of converting xylose to ethanol, including *Kluyveromyces cellobiovorus*, *Pachysolen tannophilus*, *Spathaspora passalidarum*, *Spathaspora arborariae*, *Scheffersomyces shehatae* and *Scheffersomyces stipitis*. Furthermore, the performance of the pentose-fermenting microorganisms is usually inferior to that obtained with the microorganisms that are usually used for the fermentation of hexoses, such as the *Saccharomyces cerevisiae* and *Zymomonas mobilis* species (*Talebnia, Karakashev & Angelidaki, 2010*). To make ethanol production commercially viable, an ideal microorganism should utilize a broad range of substrates, the ethanol yield, titre and productivity should be high, and it should have a high tolerance to ethanol, temperature and possible inhibitors present in the hydrolysate (*Saini, Saini & Tewari, 2015*). Nevertheless, given the existing microbial biodiversity on the planet, the occurrence of species that have not yet been identified or associated with alcoholic fermentation of pentoses and that exhibit unregistered advantages over the species recognized as xylose fermenting is likely. These still unknown microorganisms may even use metabolic strategies or carry key enzyme genes for xylose methabolism that would allow its successful industrial application or support new metabolic engineering strategies to modify industrial microorganisms already used in biorefineries. This study, therefore, embraced the isolation of naturally occurring fungi with the ability to assimilate xylose and the selection and identification of those capable of converting xylose to ethanol. The evaluation of the performance of selected strains for the production of ethanol in synthetic media containing xylose was also an object of this study.

## MATERIAL AND METHODS

### Isolation of xylose assimilator yeast

The fungi were isolated from samples of fruits and roots that included avocados (*Persea americana*), bananas (*Musa balbisiana*), potatos (*Solanum tuberosum*), beets (*Beta vulgares*

*esculenta*), taro (*Colocasia esculenta*), passion fruit (*Passiflora sp.*), pepper (*Capsicum annuun*) and tomatos (*Solanum lycopersicum*). All these biomasses were obtained at local fairs and markets and at an advanced stage of maturity or early natural microbial decomposition. Sugarcane bagasse (*Saccharum officinarum*) and sweet sorghum bagasse (*Sorghum bicolor* L. Moench) were also used as sources of microorganism samples.

Microorganisms of interest were isolated from 5-g portions of the previously fragmented plant sample that were transferred to conical flasks containing 50 mL of YNBX medium (0.67% yeast nitrogen base and 3% D-xylose) with 0.02% chloramphenicol (*Lorliam et al., 2013*). These flasks were incubated at 28 °C for 120 h with stirring at 150 rpm in an orbital incubator (Nova Ética model 430) for prior enrichment of the population of fungi that assimilate D-xylose. Every 24 h, 100-μL aliquots of culture medium were collected, inoculated by spreading over YNBX containing 1.5% agar, and incubated at 28 °C for an additional 48 h. The colonies were isolated with the aid of a platinum loop, suspended in sterile water, inoculated in solid YMPD medium (0.3% yeast extract, 0.3% malt extract, 0.5% peptone, 1% glucose and 1.5% agar), and incubated for 48 h at 28 °C to confirm the purity of the colonies. The isolated and purified colonies were inoculated in liquid YMPD medium and incubated for 48 h at 28 °C. Sufficient glycerol was added to the medium to furnish a 10% solutions, 1-mL aliquots were transferred to cryogenic tubes, and the pure cultures were stored at −18 ± 1 °C for subsequent tests of fermentability, characterization and identification.

## Assay of gas production from xylose as sole carbon source

The ability of the isolates to produce gas in the presence of xylose as the sole source of carbon was evaluated in test tubes with screw caps containing inverted Durhan tubes and the YNBX medium, according to the procedure described by *Kurtzman (2011)*. The experiment was conducted at 28 °C in an orbital shaker at 120 rpm for 21 days with daily monitoring of gas production. The CBS6054 lineage of *Scheffersomyces stipitis* was used as a positive control.

## Morphological and biochemical characterization of the selected fungi

Macroscopic and microscopic observations of the selected gas producing strains were performed after growth on solid YMPD medium at 28 °C for 48 h for the morphological characterization. In the macroscopic observations were analysed the texture, color, shape, type of surface, border and profile characteristics with aid of a stereoscope at 40× magnification. Microscopic examination was performed using 400× and 1,000× magnification with the aid of a trinocular microscope coupled to a 5.0-Mpixel digital camera to capture the images. The formation of pseudohyphae and true septated hyphae and the cell shapes were evaluated.

The reactivity of the colonies with Diazonium B Blue (DBB) was used to distinguish between ascomycetes and basidiomycetes (*Hagler & Ahearn, 1981*). The effect of temperature (28 °C, 30 °C, 35 °C, 37 °C, 40 °C and 42 °C) on the growth of the selected strains in YMPD medium was also observed for a period of up to 21 days. Biochemical assays of the assimilation of different nitrogen and carbon sources (cadaverine, creatinine, nitrate,

nitrite, lysine, glucose, maltose, raffinose, trehalose, xylose, arabinose, fructose, sucrose, inulin, meso-erythrytol, methanol, xylitol, glycerol, starch, melibiose and galactose) were performed according to the procedure described by *Kurtzman (2011)*. The technique of replica plating on solid media containing basal agar (0.67% YNB, 2% agar) and 0.2% of carbon or nitrogen source was used. The growth of colonies was observed for up to 48 h. The biochemical assay to verify the fermentation of sugars (glucose, xylose, sucrose, fructose, maltose, raffinose, galactose, melibiose and ribose) was also used to characterize the isolates utilizing the procedure described by *Kurtzman (2011)*. To 2 mL of basal medium (4.5 g yeast extract, 7.5 g of peptone and 3 mg of bromothymol blue in 1 L of distilled water), previously autoclaved in test tubes with screw caps and containing inverted Durham tubes, was added 1 mL of 6% sugar solution, and the solution was inoculated with 100 μL of microbial suspension. The tubes were kept at 28 °C and monitored for 21 days for the production of gas. The ability of the selected strains to hydrolyze starch and to produce urease was also investigated according to the method described by *Kurtzman (2011)*.

## Molecular identification of selected fungi

Colonies grown on solid YMPD medium were transferred to centrifugal microtubes with the aid of a platinum loop, and the DNA was extracted according to the method described by *Green & Sambrook (2012)*. Estimation of the amount and quality of extracted DNA was performed by electrophoresis in 1% agarose gel (w/v) followed by DNA band revelation with ethidium bromide. The molecular identification of the selected strains was performed by sequencing of rDNA regions using the NL1 (5′-GCATATCAATAAGCGGAGGAA-3′) and NL4 (5′-GGTCCGTGTTTCAAGACGG-3′) primers for amplification of the D1/D2 domain of the gene responsible for encoding the 26S rRNA region. The ITS1 (5′-TCCGTAGGTGAACCTGCGG-3′) and ITS4 (5′-TCCTCCGCTTATTGATATGC-3′) primers were employed for the amplification of the gene responsible for encoding the 5.8 S rRNA region. Amplification of regions of interest and sequencing of the PCR products were accomplished by the Macrogen Company (Rockville, MD, USA; http://www.macrogenusa.com). The sequences obtained were compared with sequences deposited in the GenBank nucleotide database (National Center for Biotechnology Information, NCBI, http://www.ncbi.nlm.nih.gov) using the BLAST program (basic local alignment search tool) (*Zhang et al., 2000*).

## Assay of alcoholic fermentation

The isolates that tested positive for the production of gas from xylose were then evaluated for their capacity for the production of ethanol. The ethanol production assays were performed with the liquid medium described by *Oliveira (2010)*, contained 20 g $L^{-1}$ of xylose, 1.25 g $L^{-1}$ of urea, 1.1 g $L^{-1}$ of $KH_2PO_4$, 2 g $L^{-1}$ of yeast extract and 40 mL $L^{-1}$ of micronutrient solution (12.5 g $L^{-1}$ $MgSO_4.7H_2O$, 1.25 g $L^{-1}$ $CaCl_2$, 2.5 g $L^{-1}$ citric acid, 10.9 g $L^{-1}$ $FeSO_4.7H_2O$, 0.19 g $L^{-1}$ $MnSO_4$, 0.3 g $L^{-1}$ $ZnSO_4.7H_2O$, 0.025 g $L^{-1}$ $CuSO_4.5H_2O$, 0.025 g $L^{-1}$ $CoCl_2.6H_2O$, 0.035 g $L^{-1}$ $(NH_4)_6MoO_{24}.4H_2O$, 0.05 g $L^{-1}$ $H_3BO_3$, 0.009 g $L^{-1}$ KI, and 0.0125 g $L^{-1}$ $Al_2(SO_4)_3$). The final pH was 5.0.

The frozen stock cultures were reactivated in YMPD solid medium and inoculated in 250-mL conical flasks containing 150 mL of previously described liquid medium,

followed by incubation at 28 °C on an orbital shaker at 150 rpm until an optical density of one unit at 610 nm was reached. Subsequently, 20 mL of this culture containing thickened yeast was used as an inoculum for evaluation of ethanol production in the same medium. Fermentation experiments were conducted in conical 250-mL flasks with hydrophobic cotton plugs. The flasks contained 80 mL of medium, to which was added 20 mL of inoculum, were incubated at 28 °C on an orbital shaker at 150 rpm for 72 h. The fermentation process was monitored every 12 h by means of the determination of the concentration of reducing sugars, by a colorimetric method (*Miller, 1959*), and monitoring cell growth profile using a Neubauer chamber. The cell growth profile also was expressed in dry weigth by means of experimental correlation with cell suspension optical density reading at 610 nm.

At the end of each fermentation, the concentration of ethanol was determined using a spectrophotometric method (*Isarankura-Na-Ayudhya et al., 2007*). The production of ethanol was confirmed by HPLC, as described by *Matos et al. (2018)*. The product yield by substrate ($Y_{P/S}$, g g$^{-1}$) was calculated as the ratio between the amount of ethanol produced and the amount of reducing sugars consumed at end of fermentative process; the ethanol yield by cell growth ($Y_{P/X}$, g g$^{-1}$) was calculated as the ratio between the amount of ethanol produced and the amount of cell biomass formed at end of fermentative process; the fermentation efficiency (Ef%) were calculated as ratio between $Y_{P/S}$ and theoretical alcoholic fermentative yield (0.511); the volumetric productivity ($Q_P$, g L$^{-1}$ h$^{-1}$), which is the ratio between the final concentration of ethanol and the fermentation time, also was acessed. The specific growth rate ($\mu X$, h$^{-1}$) was calculated as the number of cells produced in a defined time interval during the exponential growth phase using $\mu X = \frac{\ln(X_1/X_0)}{t_1 - t_0}$, where $x_1$ and $x_0$ are the cellular concentrations (dry weigth) and $t_1$ and $t_0$ are the culture times (hours). The alcoholic fermentation tests and all the analytical determinations were performed in triplicate. Tukey's test was performed at 0.05 *p*-level for comparison of means.

## RESULTS

### Isolation and selection of xylose fermenting fungi

Two hundred and two microbial colonies that were able to grow in solid medium with xylose as the sole carbon source were isolated from different ten plant biomasses (Table 1). However, only three microbial isolates, coded as UFVJM-R10, UFVJM-R150 and UFVJM-R131, which were obtained from taro, tomato and sugarcane bagasse samples, respectively, were able to produce gas in liquid medium containing xylose as the sole carbon source. This production occurred within 96 h of cultivation. Gas production was interpreted as evidence that these isolates could eventually achieve the desired alcoholic fermentation of xylose, which produces $CO_2$ as a coproduct. The confirmation of this expectation was evaluated with fermentative tests followed by determination of the production of ethanol.

### Morphological characterization of isolated fungi strains

Photographs of the morphotypes of the D-xylose-fermenting strains cultured in solid YMPD medium at 28 °C for 48 h can be observed in Figs. 1 and 2. All the colonies had

**Table 1** Number of colonies isolated from plant biomass and capable of growing in a culture medium containing xylose as the sole carbon source.

| Origin of isolated colonies | Isolated colonies | |
|---|---|---|
| | No | % |
| **Avocado** (*Persea americana*) | 20 | 9.9 |
| **Sugarcane bagasse** (*Saccharum officinarum*) | 18 | 8.9 |
| **Saccharine sorghum bagasse** (*Sorghum bicolor*) | 20 | 9.9 |
| **Banana** (*Musa balbisiana*) | 23 | 11.4 |
| **Potato** (*Solanum tuberosum*) | 23 | 11.4 |
| **Beet** (*Beta vulgares esculenta*) | 18 | 8.9 |
| **Taro** (*Colocasia esculenta*) | 20 | 9.9 |
| **Passion fruit** (*Passiflora edulis*) | 21 | 10.4 |
| **Pepper** (*Capsicum annuun*) | 18 | 8.9 |
| **Tomato** (*Solanum lycopersicum*) | 21 | 10.4 |
| **Total isolated colonies** | 202 | 100 |

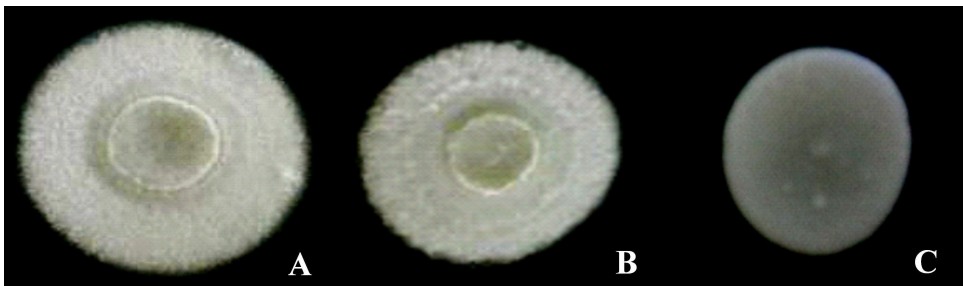

**Figure 1** Macroscopic appearance of colonies of yeast strains selected as D-xylose fermenting: UFVJM-R10 (A), UFVJM-R150 (B) and UFVJM-R131 (C) isolated from taro, tomato and sugarcane bagasse samples, respectively (40× magnification).

circular shapes and a lack of diffuse pigment (Fig. 1). The UFVJM-R10 and UFVJM-R150 strains had surfaces with concentric grooves, a border, filamentous growth and a radial aspect. A smooth profile and a central concavity in the form of a crater were also observed in the UFVJM-R10 and UFVJM-R150 strains. A creamy appearance, smooth surface and edges, flat profile and yellowish-white color were observed for the UFVJM-R131 strain.

The growth of the selected strains cultivated in solid YMPD medium at 28 °C was also evaluated (Table 2). All the selected strains had a minimum radial growth of 7 mm after 24 h of culture. The colonies of the UFVJM-R10 and UFVJM-R150 strains doubled in size in 48 h. Considering the time interval between 24 and 336 h, the size of the colonies of these same strains increased about eigth times, reaching up to 90 mm in diameter. The UFVJM-R131 strain increased comparatively slowly, reaching a maximum of 15 mm at the end of 336 h of cultivation.

As for the microscopic appearance of strains grown in YMPD at 28 °C for 48 h, cylindrical cells formed by true mycelium hyphae, positive germ tube and the presence of ascospores, chlamydospores and arthroconidia (Fig. 2) were observed for the the UFVJM-R10 and

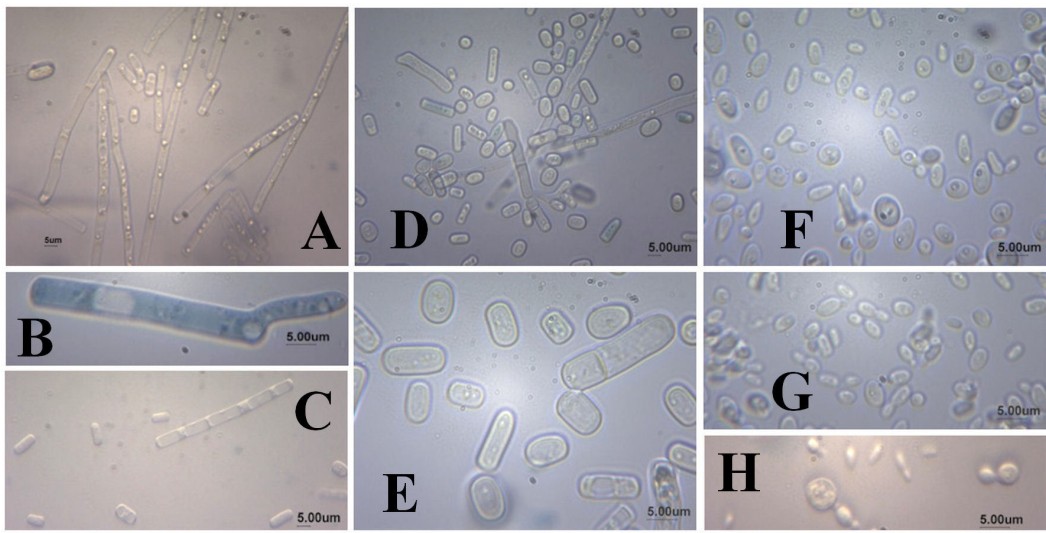

**Figure 2** Microscopical appearance of fungal strains having the ability to ferment D-xylose: UFVJM-R10 (A, B, C), UFVJM-R150 (D, E) and UFVJM-R131 (F, G, H). The A, C and D images magnified 400× and other images with 1,000× magnification are shown. Scale bar = 5 μm.

**Table 2** Size of the colonies of the UFVJM-R10, UFVJM-R150 and UFVJM-R131 strains isolated from taro, tomato and sugarcane bagasse, respectively, as a function of growth time in YMPD medium at 28 °C.

| Time (h) | Colony Diameter (mm) | | |
|---|---|---|---|
| | UFVJM-R10 | UFVJM-R150 | UFVJM-R131 |
| 24 | 12 | 11 | 7 |
| 48 | 21 | 20 | 9 |
| 72 | 30 | 30 | 10 |
| 168 | 50 | 68 | 14 |
| 336 | 90 | 90 | 15 |

UFVJM-R150 strains. Globular and ovoid cells, pseudo-hyphae formation, and the presence of ascospores and blastoconidia were observed for the UFVJM-R131 strain.

The selected strains grew at 28 °C and 30 °C after 24 h of cultivation in YMPD medium, but no growth was observed at temperatures equal to or higher than 35 °C. This fact characterizes them as mesophilic. The DBB test realized with isolated colonies was negative for all the three selected strains, thereby indicating that they belong to the Ascomycetes group (*Hagler & Ahearn, 1981*). The results of the tests for starch hydrolysis and production of urease were negative for all three strains. These selected fungal strains possessed the ability to assimilate the pentoses D-xylose and L-arabinose (Table 3). With the exception of meso-erythritol and inulin, all other carbon sources tested were assimilated within 48 h. The UFVJM-R10 and UFVJM-R150 strains did not assimilate inulin. All the nitrogen sources tested (cadaverine, creatinine, nitrate, nitrite and L-lysine) were assimilated by the three strains (Table 3). The selected strains also exhibited the capacity to ferment

**Table 3  Biochemical assimilation of carbon and nitrogen sources by selected D-xylose-fermenting fungi strains.**

| Carbon and nitrogen sources | Strains | | |
|---|---|---|---|
| | UFVJM-R10 | UFVJM-R150 | UFVJM-R131 |
| Cadaverine | +[a] | +[a] | +[a] |
| Creatinine | +[a] | +[a] | +[a] |
| Nitrate | +[a] | +[a] | +[a] |
| Nitrite | +[b] | +[a] | +[b] |
| L-lisine | +[b] | +[a] | +[a] |
| Glucose | +[a] | +[a] | +[a] |
| Maltose | +[a] | +[a] | +[a] |
| Sucrose | +[a] | +[a] | +[a] |
| Fructose | +[a] | +[a] | +[a] |
| L-Arabinose | +[b] | +[b] | +[a] |
| D-Xylose | +[a] | +[a] | +[a] |
| Galactose | +[a] | +[a] | +[a] |
| Melibiose | +[a] | +[a] | +[a] |
| Meso-erythritol | − | − | − |
| Trehalose | +[a] | +[a] | +[a] |
| Raffinose | +[a] | +[a] | +[a] |
| Inulin | − | − | +[b] |
| Glycerol | +[a] | +[a] | +[a] |
| Methanol | +[a] | +[a] | +[a] |
| Starch | +[a] | +[a] | +[a] |

**Notes.**

Growth: positive (+) and negative (−).

[a]Time of growth: 24 h.

[b]48 h.

glucose, fructose and xylose (Table 4). None of the strains was able to ferment D-ribose or melibiose. The UFVJM-R131 strain was the only one capable of fermenting sucrose, maltose, galactose and raffinose. The results of fermentation, when positive, were observed within 48 h of incubation.

## Homology search of 5.8S rDNA and 26S rDNA regions

Nucleotide sequences of the PCR products obtained by amplifying the regions of the small and large subunit ribosomal RNA genes from selected xylose-fermenting strains were deposited on GenBank-NCBI (Table 5). The degree of identity in the NCBI nucleotide database was searched using the BLAST tool (*Zhang et al., 2000*). The UFVJM-R10 isolate presented 99% identity to strains of *Geotrichum candidum* and *Galactomyces geotrichum* when the D1/D2 region of the 26S rDNA amplified sequence was researched (GenBank Accession Number MF362099) (Table 5). A 100% identity with strains of *Geotrichum candidum* and *Galactomyces geotrichum* was observed for the UFVJM-R150 isolate using the ITS region amplified with ITS1/ITS4 primers (GenBank Accession Number MF360015) as a parameter for comparison to the GenBank. A 99% identity with the *Galactomyces geotrichum* and *Geotrichum candidum* species was observed for the same microbial isolate

**Table 4  Results of carbohydrate fermentation tests for the strains selected as D-xylose-fermenting.**

| Carbohydrates | Strains | | |
|---|---|---|---|
| | UFVJM-R10 | UFVJM-R150 | UFVJM-R131 |
| Glucose | $+^a$ | $+^a$ | $+^a$ |
| Sucrose | − | − | $+^a$ |
| Maltose | − | − | $+^a$ |
| D-ribose | − | − | − |
| Galactose | − | − | $+^b$ |
| Fructose | $+^b$ | $+^b$ | $+^a$ |
| Xylose | $+^b$ | $+^b$ | $+^b$ |
| Melibiose | − | − | − |
| Raffinose | − | − | $+^a$ |

**Notes.**
Fermentation: positive (+) and negative (−).
[a]Time of fermentation: 24 h.
[b]48 h.

**Table 5  Molecular identification of selected D-xylose-fermenting fungi.**

| Strain | Accession number | Identity based on ITS 5.8S rDNA sequence | Identity based on D1/D2 26S rDNA sequence |
|---|---|---|---|
| UFVJM-R10 | MF362099[a] | ND | *Galactomyces geotrichum* (99%) *Geotrichum candidum* (99%) |
| UFVJM-R150 | MF360015[a] MF371338[b] | *Geotrichum candidum* (100%) *Galactomyces geotrichum* (100%) | *Galactomyces geotrichum* (99%) *Geotrichum candidum* (99%) |
| UFVJM-R131 | KY325443[a] KY325444[b] | *Candida akabanensis* (98%) | *Candida akabanensis* (99%) |

**Notes.**
[a]GenBank accession number of ITS 5.8S rDNA sequence.
[b]GenBank accession number of D1/D2 26S rDNA sequence.
ND, Not Determined.

when the partial sequence of the D1/D2 region of the 26S rDNA amplified with NL1/NL4 primers (GenBank accession number MF371338) was researched. A 98% identity with *Candida akabanensis* species was observed for the UFVJM-R131 isolate when the ITS region amplified with ITS1/ITS4 primers (GenBank accession number KY325443) was used as a reference. A 99% identity with the same species was observed when the partial sequence of the D1/D2 region of the 26S rDNA gene amplified with the NL1/NL4 primers (GenBank accession number KY325444) was used as a reference in the GenBank.

## Production of ethanol by selected isolates

The *G. geotrichum* UFVJM-R10, *G. geotrichum* UFVJM-R150 and *C. akabanensis* UFVJM-R131 strains were evaluated with regard to the consumption of xylose for microbial growth and for the production of ethanol. The three fungi strains consumed 100% of the xylose available in 60 h (Fig. 3). Considering only the growth curves expressed in dry weight, the *G. geotrichum* UFVJM-R10 and *G. geotrichum* UFVJM-R150 strains were shown to exhibit

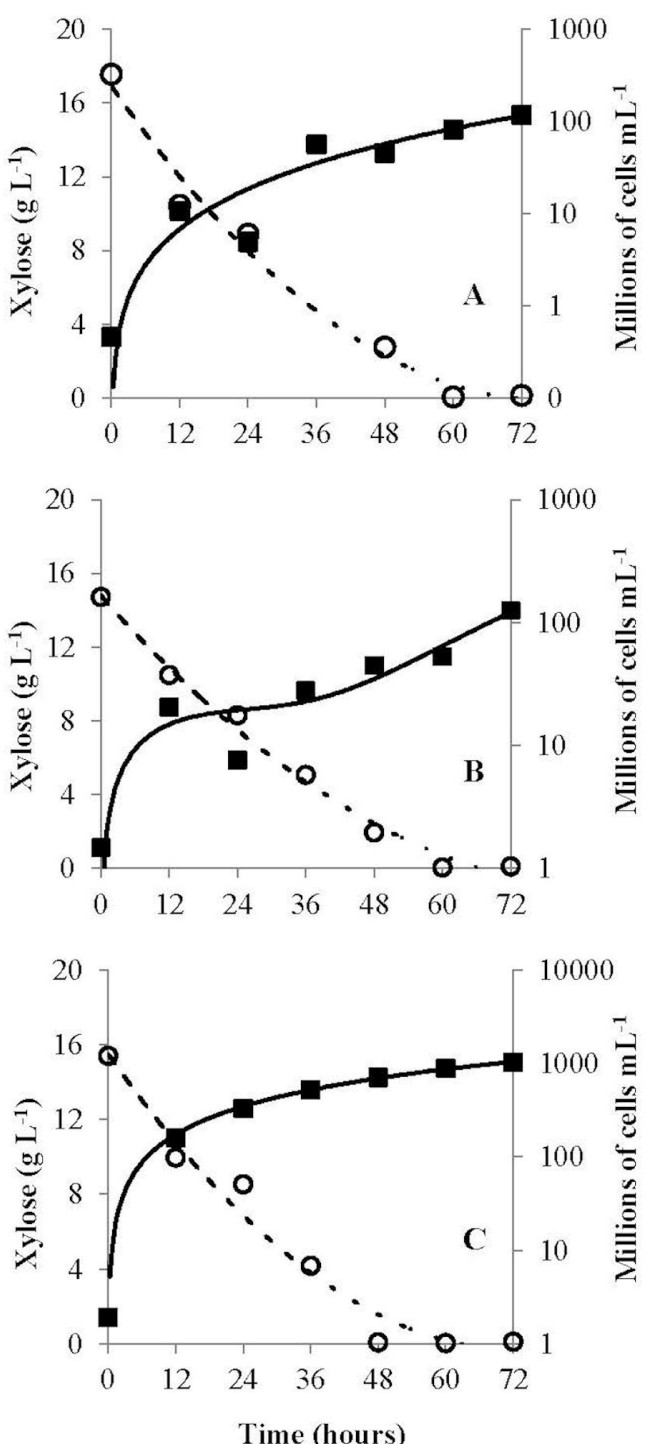

**Figure 3** Progress curve (cell growth *vs* carbohydrate consumption) of *G. geotrichum* UFVJM-R10 (A), *G. geotrichum* UFVJM-R150 (B) and *C. akabanensis* UFVJM-R131 (C). Open circle, xylose concentration; full square, cell growth.

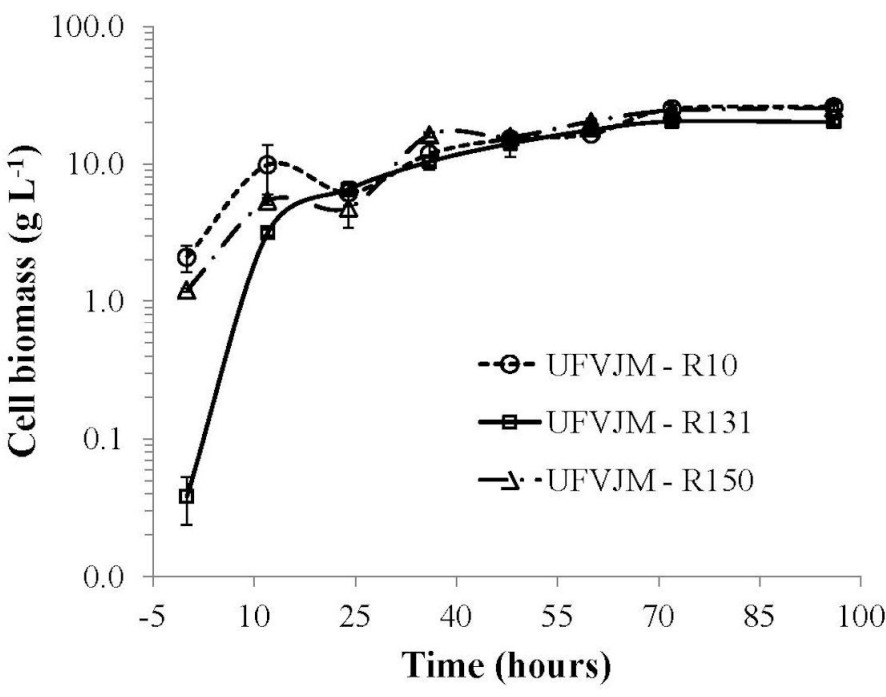

**Figure 4** Cell growth pattern of three selected D-xylose fermenting strains expressed in dry weight.

**Table 6 Response variables for the fermentative performance of xylose-fermenting yeast.**

| Fungi strain | $Y_{P/S}$ ($g_p$ $g_s^{-1}$) | $Y_{P/X}$ ($g_p$ $g_x^{-1}$) | E$f$ (%) | Ethanol (g L$^{-1}$) | $Q_P$ (g L$^{-1}$h$^{-1}$) | $\mu X$ (h$^{-1}$) |
|---|---|---|---|---|---|---|
| *G. geotrichum* UFVJM-R10 | 0.35[a] | 0.32[a] | 69.4 | 5.03[a] | 0.07[a] | 0.13[a] |
| *G. geotrichum* UFVJM-R150 | 0.29[a] | 0.23[a] | 57.0 | 5.05[a] | 0.07[a] | 0.12[a] |
| *C. akabanensis* UFVJM-R131 | 0.34[a] | 0.25[a] | 66.7 | 5.12[a] | 0.07[a] | 0.37[b] |

The presence of the same letter next to the data of the same column indicates that there was no statistical difference among the averages recorded with the Tukey's test ($p = 0.05$).

the same growth profile (Fig. 4). The steady state growth was achieved within 72 h after initiating the process (Fig. 4).

The yields resulting from the fermentation as a function of the substrate consumption ($Y_{P/S}$) and cell growth ($Y_{P/X}$), fermentation efficiency (Ef), volumetric productivity ($Q_P$), the production of ethanol and specific growth rate ($\mu$) are presented in Table 6. The values of $Y_{P/S}$ ranged from 0.29 to 0.35, but they were not statistically different from one another. The ethanol production reached 5.12 g L$^{-1}$. However, there were no significant differences between the $Q_P$ values or ethanol production for any of the strains. The specific growth rates observed for the *C. akabanensis* UFVJM-R131 strain (0.37 h$^{-1}$) was at least three times greater than those calculated for the other two strains (Table 6).

## DISCUSSION

The ability for alcoholic fermentation of xylose was observed to be unusual among isolated microorganisms, as expected (*Kuhad et al., 2011*). Only three (1.5%) of the two hundred
and two microbial isolates capable to grow on xylose were able to produce ethanol from this same carbon source. From a strictly morphological point of view, UFVJM-R10 and UFVJM-R150 isolates grew with cell structures containing true hyphae, whereas the UFVJM-R131 isolate had a predominantly globular or ovoid unicelular structure (Fig. 2). This difference in recorded cellular structures restricts direct comparison of the growth profiles between these different species when quantified on the basis of cell count (Fig. 3). However this limitation was bypassed when the cell growths were expressed in dry weight (Fig. 4). The UFVJM-R10 and UFVJM-R150 strains presented the same morphological and biochemical characteristics evaluated in this study (Figs. 1 and 2, Tables 3 and 4), including the growth profile (Fig. 4), and presented 99% homology to each other when the sequences of the amplified D1/D2 region were compared. Both the 26S rDNA regions amplified from the UFVJM-R10 and UFVJM-R150 strains and the 5.8S rDNA regions amplified from the UFVJM-R150 strain presented a probable identity with those of the *Galactomyces geotrichum* and *Geotrichum candidum* species when search with BLAST (Table 5). This is not a coincidence since *Galactomyces geotrichum* was considered to be the teleomorphic state of *Geotrichum candidum* until 2004. After that date, as the result of a taxonomic revision of some species of the *Geotrichum* gender (*De Hoog & Smith, 2004*; *Pottier et al., 2008*), *Geotrichum candidum* began to be considered as the anamorphic state of *Galactomyces candidus,* and *Galactomyces geotrichum* came to be considered as the teleomorphic status of an still unnamed species of *Geotrichum*. Nevertheless, the ability of *G. candidus* to grow at 35 °C (*Kurtzman, 2011*) is a characteristic that was not observed for the isolated strains studied here. Therefore, it is likely that the UFVJM-R10 and UFVJM-R150 strains, whose identity is associated with *Geotrichum candidum* or *Galactomyces geotrichum* by nucleotide homology, appear to be the *G. geotrichum* species. According to the taxonomic description proposed by *De Hoog & Smith (2011)*, species of the *Galactomyces* genus was described as white colonies, flour-like or filamentous, usually with true hyphae. The colonies have a dry aspect and radial growth when grown in medium containing glucose, peptone and yeast extract. All these characteristics were observed for the UFVJM-R10 and UFVJM-R150 strains. The positive test for assimilation and xylose fermentation was too consistent with that expected for *G. geotrichum*. Other evidence of the association of the identity of the UFVJM-R10 and UFVJM-R150 strains with the *Galactomyces geotrichum* species was described by *Moret & Sperti (1962)*, who observed the capacity of *Geotrichum candidum* to reduce xylose to xylitol and to oxidize xylitol to xylulose, thereby integrating it with the metabolism via the pentose phosphate pathway. At the time of publication of Moreti and Sperti's work, *Geotrichum candidum* was still considered to be the anamorph of the *Galactomyces geotrichum*.

Very few references exist regarding ethanol production from xylose by fungi of the *Geotrichum* genus. *Lorliam et al. (2013)* obtained 0.11 g L$^{-1}$ of ethanol from medium containing 6% xylose and inoculated with an isolate identified as *Geotrichum sp*. *Nigam et al. (1985)* have isolated four strains of *Geotrichum sp*. that are capable of producing more than 1 g of ethanol per liter in a medium containing 2% xylose. The results for ethanol production by the *G. geotrichum* strains evaluated in the present study were at least five times higher than those obtained by those authors.

The molecular identification of the amplified nucleotide sequence of UFVJM-R131 in the GenBank only returned *C. akabanensis* as the probable species, and this result drew support from the positive results obtained with the biochemical fermentation assays of sucrose, galactose and raffinose (see Table 4) (*Kurtzman, 2011*). The positive result with inulin in the assimilation test was also significant for the characterization of this species (*Kurtzman, 2011*). *C. akabanensis* had already been used for alcoholic fermentation of the Agave leaf juice, which is rich in sucrose, fructose and glucose, and an efficiency of 88% for the production of ethanol was obtained (*Corbin et al., 2016*). However, the potential for fermentation of xylose was not assessed by the authors because this sugar was not identified in the juice from the *Agave tequilana* leaf.

In a recently published work by our research group (*Matos et al., 2018*), the microbial isolates described herein were able to ferment the hemicellulosic hydrolyzate obtained by acid treatment of the sunflower seed cake. In that work, isolates UFVJM-R10 and UFVJM-R131 presented $Y_{P/S}$ values of 0.29 and 0.27 g ethanol $g^{-1}$ sugars, respectively, and were capable to cofermenting xylose and glucose.

## CONCLUSION

Three new strains of yeast capable of converting xylose into ethanol were isolated and identified. Two of the strains were identified as belonging to *Galactomyces geotrichum* species. Another strain was identified as *Candida akabanensis*. This report leaves room for the study and application of these species for the production of lignocellulosic ethanol.

## ACKNOWLEDGEMENTS

The authors thank the Foundation André Tosello (SP, Brazil) for the donation of the *Scheffersomyces (Pichia) stipitis* CBS6054 strain.

### Funding

This work was supported by Fundação de Amparo a Pesquisa do Estado de Minas Gerais (FAPEMIG) and by the Conselho Nacional de Desenvolvimento Científico e Tecnológico (CNPq/Brazil). DL Nelson was the recipient of a PVNS fellowship from the Coordenação de Aperfeiçoamento de Pessoal de Ensino Superior (CAPES/Brazil). The funders had no role in study design, data collection and analysis, decision to publish, or preparation of the manuscript.

### Grant Disclosures

The following grant information was disclosed by the authors:
Fundação de Amparo a Pesquisa do Estado de Minas Gerais (FAPEMIG).
Conselho Nacional de Desenvolvimento Científico e Tecnológico (CNPq/Brazil).
Coordenação de Aperfeiçoamento de Pessoal de Ensino Superior (CAPES/Brazil).

## Competing Interests

The authors declare there are no competing interests.

## Author Contributions

- Raquel V. Valinhas performed the experiments, analyzed the data, prepared figures and/or tables, authored or reviewed drafts of the paper, approved the final draft.
- Lílian A. Pantoja conceived and designed the experiments, analyzed the data, authored or reviewed drafts of the paper, approved the final draft.
- Ana Carolina F. Maia performed the experiments, authored or reviewed drafts of the paper, approved the final draft.
- Maria Gabriela C.P. Miguel performed the experiments, contributed reagents/materials/analysis tools, authored or reviewed drafts of the paper, approved the final draft.
- Ana Paula F.C. Vanzela analyzed the data, contributed reagents/materials/analysis tools, authored or reviewed drafts of the paper, approved the final draft.
- David L. Nelson prepared figures and/or tables, authored or reviewed drafts of the paper, approved the final draft, english translation.
- Alexandre S. Santos conceived and designed the experiments, performed the experiments, analyzed the data, contributed reagents/materials/analysis tools, prepared figures and/or tables, authored or reviewed drafts of the paper, approved the final draft.

## DNA Deposition

The following information was supplied regarding the deposition of DNA sequences:
GenBank: MF362099, MF360015, MF371338, KY325443, KY325444.

## Supplemental Information

Supplemental information for this article can be found online at http://dx.doi.org/10.7717/peerj.4673#supplemental-information.

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
