# Peer review of "Xylose fermentation to ethanol by new Galactomyces geotrichum and Candida akabanensis strains"

_PeerJ, doi:10.7717/peerj.4673_

## Round 0.1 · original submission · Minor Revisions

One of the reviewers recommended major revision and the other recommended minor revision. All in all, I think that responding to the reviewers' comments should be sufficient for publication of this manuscript in PeerJ.

Reviewer 1 ·

Basic reporting

No comment

Experimental design

No comment.

Validity of the findings

No comments.

Additional comments

This manuscript describes the isolation, identification and characterization of new xylose-fermenting yeast. The work is interesting and significant amount of work has been done. However, the current version is not publishable if the following minor comments are addressed?
Minor comments
• Besides ethanol production, naturally occurring xylose-fermenting normally produced significant amount of xylitol. Authors should include xylitol production in characterization as majority xylose-utilizing yeast primarily produce xylitol.
• Yp/p = 0.29 – 0.35 g/g were quite significant. It is also interesting to know the yield of xylitol and cell biomass.
• Glucose and xylose are most abundant monomer sugars in lignocellulosic biomass hydrolysate. Could authors also characterize the strains on glucose and xylose co-utilization and co-fermentation?
• Currently there are many high-performance engineered xylose-fermenting S. cerevisiae. They not only could ferment xylose, but also able to co-ferment glucose xylose and produce high-concentration ethanol. In addition, they are also tolerant to the inhibitors derived from biomass hydrolysate. Are there any advantages in using the naturally occurring xylose-fermenting yeast over the metabolically engineered S. cerevisiae? Although discovery of new naturally occurring xylose-fermenting yeast is interesting, we cannot ignore the fact that plenty of recombinant xylose-fermenting yeast were developed. They are probably more suitable for industrial application. Author should include this in introduction and results/discussion.
• Line 50, $ sign at the beginning sentence should be deleted.
• Line 273-274: The ethanol production reached 5.12 g L-1. This was very low. What if sugar concentration is high? Can the selected strains tolerant ethanol?
• Line 312: should be “was described”.

Reviewer 2 ·

Basic reporting

I found a clear report of the data and supportive of the findings the authors commented on the text.
Some literature citations are not well cited (please check general comments to the authors).
And the discussion requires a better job to connect relevant previous findings of the bioethanol production field.

Experimental design

The experimental design is correct, but it needs some description about the calculation of some parameters (please check general comments).

Validity of the findings

The results provided are novel and interesting for further research.

Additional comments

The isolation of new strains with the capacity of ethanol production from xylose fermentation is an important source of new genetic traits for improving the ethanol yields of the bioethanol industry.
Here, Valinhas et al. isolated and identified three of two hundred and two yeast strains from plant samples, which are able to produce ethanol from xylose consumption. They characterized the ethanol productivity in synthetic media with xylose supporting previous results where they demonstrated that the production of ethanol from sunflower hydrolysate, using 2 of the 3 strains explored in this paper, is due to xylose and not only from other sugar consumption (Matos et al 2018 Quim. Nova).
I found an interesting piece of work, which might be published in PeerJ if some of the issues, I reported below, are fixed.

Major comments:
1. I missed in the discussion a comparison of the new strains with industrial xylose fermenting strains, and how the new strains identified by the authors might replace the industrial ones, which I consider from the data presented unlikely, or they must explain how the new strains might improve the existing ones.
2. Authors consider UFVJM-R10 and UFVJM-R150 to show the same growth profile a wrong conclusion based on the shape of the presented curves in Figure 4. While R10 showed two curves probably due to a shift in its metabolism before finishing the fermentation, R150 has not that behavior. Please correct that conclusion.
Minor comments:
1. Authors provided supplementary data that I do not see a citation in the text. In addition, the supplementary data must be translated to English.
2. How are the maximum growth rate calculated? There is not citation to the methodology nor explanatory text.
3. Check references. In many places, for example Matos et al in lane 171 and in the discussion lane 339, there are incorrect number correspondence to the citation.
4. In table 4 there are conditions with identical response in all three strains. It must be removed based on the table description title.
Minor corrections
1. A dollar symbol is found in lane 51.
2. Typo in lane 58: change ethano to ethanol
3. Typo in lane 73: change Methodes to methods.
4. Typo in lane 171: change comfirmed to confirmed.
5. Typo in lane 180 and in other parts of the text: change Two hundred two to Two hundred and two.
6. Through the text the authors wrote that one of the strains was isolated from taro, but in Table 2 it says yam.
7. Lane 312: Galactomyces must be in italics.

---

## Round 0.2 · accepted · Accept

Note the Minor correction from Reviewer 2 ("Authors indicated, “was describe”, line 331, as corrected but it is still showing the typo.") which can be addressed while in Production.

# Reviewer 2 ·

Basic reporting

no comment

Experimental design

no comment

Validity of the findings

no comment

Additional comments

Authors have covered all my comments and with a minor correction, I consider the revised version of “Xylose fermentation to ethanol by new Galactomyces geotrichum and Candida akabanensis strains” by Valinhas et al able to be published by PeerJ.
Minor corrections
1. Authors indicated, “was describe”, line 331, as corrected but it is still showing the typo.